

# Benchmarking of SWE products based on outcomes of the SnowPEx+ Intercomparison Project

Lawrence Mudryk[1], Colleen Mortimer[1], Chris Derksen[1], Aleksandra Elias Chereque[2], Paul Kushner[2]

[1]Climate Research Division, Environment and Climate Change Canada, Toronto, M3H 5T4, Canada
[2]Department of Physics, University of Toronto, Toronto, M5S 1A7, Canada

*Correspondence to*: Lawrence Mudryk (Lawrence.mudryk@ec.gc.ca)

**Abstract.** We assess and rank 23 gridded snow water equivalent (SWE) products by implementing a novel evaluation strategy using a new suite of reference data from two cross-validated sources and a series of product inter-comparisons. The new reference data combines in situ measurements from both snow courses and airborne gamma measurements. Compared to
previous evaluations of gridded products, we have substantially increased the spatial coverage and sample size across North America, and we are able to evaluate product performance across both mountain and non-mountain regions. The evaluation strategy we use ranks overall relative product performance while still accounting for individual differences in ability to represent SWE climatology, variability, and trends. Assessing these gridded products fills an important gap in the literature since individual gridded products are frequently chosen without prior justification as the basis for evaluating land surface and
climate model outputs, along with other climate applications. The top performing products across the range of tests performed are ERA5-Land followed by the Crocus snow model. Our evaluation indicates that accurate representation of hemispheric SWE varies tremendously across the range of products. While most products are able to represent SWE reasonably well across Northern Hemisphere non-mountainous regions, the ability to accurately represent SWE in mountain regions and to accurately represent historical trends are much more variable. Finally, we demonstrate that for the ensemble of products evaluated here,
attempts to assimilate surface snow observations and/or satellite measurements lead to a deleterious influence on regional snow mass trends, which is an important consideration for how such gridded products are produced and applied in the future.

## 1 Introduction

Historical gridded snow water equivalent (SWE) products are temporally continuous and spatially complete datasets required
across many disciplines spanning climate, hydrology, and ecology (Liston, 1999; Clark et al., 2011; Lundquist and Dettinger, 2005; Simpson et al., 2022; Orsolini et al., 2013; Dutra et al., 2011; Jones et al., 2011). Numerous such products exist based on a range of techniques: output from coupled reanalysis systems, offline simulations of snow models driven by historical meteorological forcing, and satellite-based retrievals, all of which may also assimilate snow observations from surface networks or remotely sensed data. These gridded products aim to represent various aspects of historical snow conditions (e.g.





areal coverage, surface snow amount, snow temperatures, etc.) and because of this are frequently used to evaluate terrestrial snow output from land surface and earth system models (for example, Collier et al., 2018). However, the historical gridded products themselves require validation with in situ observations.

For surface snow amounts, this validation is challenging for several reasons. In situ point snow depth measurements are the

most readily available and plentiful reference data available, however some gridded products already assimilate this data in the course of their production thereby negating its use as independent reference data. Even when not incorporated into the production of a gridded product in situ snow depths require assumptions about snow density in order to evaluate SWE and are nonideal for evaluating the spatial scale on which gridded products represent snow, which can range from roughly $10^2$-$10^4$ km$^2$. In place of point measurements, the use of snow courses/transects (WMO, 2018) is more appropriate. These provide

information on both snow density and depth to better constrain SWE, and they also represent the snowpack on a spatial scale of roughly 0.1-1 km$^2$, which is closer to the resolutions of the gridded products. Mortimer et al. (2020) previously used such data to evaluate a range of gridded products, but the analysis excluded complex terrain and had poor coverage across portions of North America. Along with snow transects, airborne gamma measurements can also be used to derive SWE estimates that are representative on similar spatial scales to those from snow transects (Carroll, 2001). These measurements compare the

attenuation of gamma radiation due to the presence of snow and compare with measurements conducted under snow-free conditions while accounting for background soil moisture. Cho et al. (2020) used historical data of this type available over the United States to evaluate a small selection of gridded products.

Recently Mortimer et al. (2024) cross-validated snow transect and airborne gamma SWE measurements over North America.

They demonstrated broad consistency in the corresponding SWE values from the two types of measurements and consistency in the relative performance of gridded SWE products as assessed using either source of reference data. These results support combining snow course and airborne gamma measurements into a single reference dataset. The result is a new suite of reference data with expanded spatial coverage and volume of measurements, thus greatly improving the validation domain across North America compared to prior studies.


We make extensive use of this new reference dataset, along with additional approaches to dataset inter-comparisons to produce what we consider the most robust and authoritative evaluation of gridded SWE products performed to date. We evaluate 24 gridded SWE products on their ability to represent aspects of SWE climatology, variability, and trends across three segments of the snow season (snow onset, seasonal peak, and snow melt) and across regions spanning the Northern Hemisphere. The

breadth of evaluation criteria permits us to make recommendations on which gridded datasets are appropriate for a variety of uses.



The sort of evaluation employed here shares philosophical connections to those employed by other projects such as ILAMB (Collier et al., 2018) and AMBER (Seiler, 2020) that aim to evaluate historical estimates of a range of land surface variables.

However, ILAMB and AMBER are concerned with multiple outputs from land surface models that are evaluated using gridded data or Fiducial Reference Measurements (which are spatially less representative of land surface model output). Our analysis is a detailed evaluation of a single variable (SWE) using both comparisons with in-situ data and gridded product inter-comparisons thereby helping to inform the reference products employed in ILAMB and AMBER. By improving the temporal continuity and spatial coverage of our analysis, our ultimate goal is to provide a validation framework that would facilitate

automated evaluation of forthcoming gridded SWE datasets.

The remainder of this paper is organized as follows. Section 2 provides the list of gridded SWE products we evaluate, outlines our overall evaluation strategy, and describes the specific evaluation metrics and range of reference data used in the evaluation. We illustrate product-specific performance over a range of tests in Sect. 3. In Sect. 4 we provide the overall product rankings

along with recommendations for which products may be used in what capacity and where their shortcomings exist (e.g. accurately captures spatial distribution of SWE, accurately captures seasonal snow mass trends, etc), along with additional discussion points and concluding remarks.

## 2 Data and Methods

### 2.1 Evaluated Gridded SWE Products

We evaluate the suite of 24 gridded SWE products listed in Table 1 (organized into families and described briefly below). While some of these products are now deprecated and have been superseded by updated versions, we include them in our evaluation as they provide a baseline to indicate the improvement or deterioration of performance with subsequent versions. Additionally, by including these older product versions, our evaluation may be useful for interpretation of previously published

analysis that used such datasets.

**Table 1** List of all gridded SWE products evaluated. The † symbol denotes products that are deprecated or superseded by updated versions. Product availability is specified in the Data availability section.

| Product Name | Abbr. | Period | Method to Estimate SWE | Surface Information |
|---|---|---|---|---|
| B-TIM-ERA5 | BE5 | 1981–2020 | SM-un / ERA5 met. | None |
| B-TIM-JRA55 | BJR | 1981–2020 | SM-un / JRA55 met. | None |
| B-TIM-MERRA2 | BM2 | 1981–2020 | SM-un / MERRA2 met. | None |
| †B-TIM-ERAint | BEI | 1981–2017 | SM-un / ERA-Interim met. | None |





| | | | | |
|---|---|---|---|---|
| Crocus-ERA5 | CE5 | 1980–2021 | SM-un / ERA5 met. | None |
| †Crocus v8 | C8 | 1980–2018 | SM-un / ERA-Interim met. | None |
| †Crocus v7 | C7 | 1080–2017 | SM-un / ERA-Interim met. | None |
| ERA5 | E5 | 1980–2020 | SM-c / ERA5 met. | SD + IMS |
| ERA5-Snow | E5S | 1980–2020 | SM-un / ERA5 met. | SD |
| ERA5-Land | E5L | 1980–2020 | SM-un / ERA5 met. | none |
| †ERA-Interim-Land | EIL | 1980–2010 | SM-un / ERA-interim met. | none |
| GLDAS v2.2 CLSM | GL22 | 2003–2018 | SM-un / Princeton met. | GRACE |
| GLDAS v2.1 Noah | GL21 | 1980–2010 | SM-un / Princeton met. | gauge precipitation |
| GLDAS v2.0 CLSM | GLc | 1980–2010 | SM-un / Princeton met. | none (open loop) |
| GLDAS v2.0 Noah | GLn | 1980–2010 | SM-un / Princeton met. | none (open loop) |
| JRA-55 | JR | 1980–2018 | SM-c / JRA55 | SD + PMW for extent |
| MERRA2 | M2 | 1980–2018 | SM-c / MERRA2 | none |
| †MERRA | M | 1980-2017 | SM-c / MERRA | none |
| JAXA-AMSR2 | JX | 2014 –2020 | Standalone PMW | none |
| Snow_CCI v2 | CC2 | 1980–2020 | PMW + SD assimilation | SD + density information |
| †Snow_CCI v1 | CC1 | 1980–2018 | PMW + SD assimilation | SD |
| GlobSnow v3 | GS3 | 1980–2018 | PMW + SD assimilation | SD |
| †GlobSnow v2.1 | GS2 | 1980–2017 | PMW + SD assimilation | SD |

**Notes:** PMW refers to SWE estimated from satellite-observations of passive microwave brightness temperatures.

SD refers to point snow depth information assimilated (data may vary by product but available sources are similar overall).

SM-c refers to coupled snow models driven by meteorological forcing as specified.

SM-un refers to uncoupled (offline) snow models driven by meteorological forcing as specified.

The Brown Temperature Index Model (B-TIM) family of products all consist of a simple temperature index snow scheme (Brown et al., 2003; Elias Chereque et al., submitted) driven by historical estimates of temperature, precipitation, and snowfall. At present, four versions of this product exist, each driven by output from a different reanalysis. The strength of these products is that they are simple to produce, require a minimal selection of driving variables, and contain no land surface assimilation so that differences among the product versions reflect differences among the driving data. This will be a key factor that we exploit in order to analyze differences in the magnitude and seasonality of regional snow mass trends among all products (Sect. 4).

The Crocus family of products are all derived from a complex snow scheme embedded in the ISBA land model (Brun et al., 2013). The most recent version is driven by ERA5 analysis fields (temperature, precipitation, humidity, winds, etc). Two



previous versions driven by fields from the now-discontinued ERA-Interim analysis are also evaluated. As will be shown in Sect. 4, these two versions have similar anomalies but differences in their parametrizations yield moderate differences in their climatologies, which affects their relative performance.

The ERA5 family of products are based on the current ECMWF reanalysis (Hersbach et al., 2023). ERA5 denotes the standard coupled reanalysis SWE output. It uses the ERA5 land surface model (HTESSEL) forced by the ERA5 meteorological analysis fields with assimilation of in situ snow depth data as available over the entire output period and IMS snow cover extent data from mid-2004 onwards. The assimilation of IMS data is known to produce a discontinuity in the climatological SWE field (Mortimer et al., 2020). To try and correct for this, ECMWF produced a second set of SWE output (denoted ERA5-Snow) using the same land surface model and forcing as the standard ERA5 product but without assimilation of IMS snow cover extent data (assimilation of snow depth data only). ERA5-Land denotes the standard uncoupled configuration of the ERA5 analysis (Muñoz Sabater, 2019) which does not assimilate any snow-related surface data. ERA-Interim-Land is the uncoupled configuration of the previous generation of the ECMWF reanalysis (Balsamo et al., 2015) and is included as a baseline product.

The GLDAS products are uncoupled configurations of the NASA Global Land Data Assimilation System Version 2. Both GLDAS-2.0 versions (Beaudoing and Rodell, 2018, 2019) are forced by the Princeton meteorological forcing input data but use two different land surface models, Catchment and Noah 3.6. The GLDAS v2.1 product (Beaudoing and Rodell, 2020b) alters the precipitation input to the Noah land surface model by incorporating information from gauge precipitation data. The GLDAS-2.2 product (Beaudoing and Rodell, 2020a) uses the CLSM land surface model and includes data assimilation of GRACE data.

The JRA-55 (JMA, 2013), MERRA2 (GMAO, 2015), and MERRA (GMAO, 2008) SWE products are standard coupled output from each reanalysis center.

We also assess five gridded products that incorporate information from passive-microwave brightness temperatures in order to fully or partially constrain surface SWE. The JAXA-AMSR2 product is a standalone passive microwave product that estimates SWE using a retrieval algorithm based only on time varying microwave brightness temperatures and other time-invariant ancillary data (Kelly et al., 2019). The remaining four Earth Observation (EO) products (GlobSnow v2 and v3 and Snow_cci v1 and v2) are related with a shared development history and their SWE output has strong similarities to one another (hereafter we refer to them collectively as GS/CCI products). All GS/CCI products use a weighted combination of passive-microwave brightness temperatures and in situ snow depth measurements to constrain SWE (Luojus et al., 2021); differences among them are detailed in (Mortimer et al., 2022).



## 2.2 Overall evaluation strategy

Products are evaluated on their ability to represent aspects of SWE climatology, variability, and trends across combinations of regions, and seasons as summarized in Table 2. SWE climatology is evaluated for two regions, SWE variability is assessed for three regions across three seasonal periods (nine combinations), and SWE trends are examined for two variables over two

seasonal periods. The choices of region and season are based on characteristics of the reference data discussed below in subsect. 2.3, while the overall number of evaluations loosely reflects the relative importance of ensuring a product has accurate variability and trends compared to its ability to reproduce the climatological SWE field.

**Table 2** Summary of evaluations performed by region, evaluation method and reference products used.

Regional abbreviations: Northern Hemisphere nonmountainous (NHnon), Eurasia nonmountainous (EAnon) North America nonmountainous (NHnon), and North America mountainous (NAm).

| Evaluation Type | Regions Tested | Method | Reference Product |
|---|---|---|---|
| Climatological SWE near seasonal peak (March) | NHnon NAm | Skill Score | Bias-corrected GlobSnow v3 Combined Snow Course + Gamma SWE (calculated mean) |
| SWE variability near seasonal peak (Feb-Mar) | EAnon NAnon NAm | Skill Score | Combined Snow Course + Gamma SWE |
| SWE variability during onset season (Sep-Jan as available) | EAnon NAnon NAm | Skill Score | Combined Snow Course + Gamma SWE |
| SWE variability during melt season (Apr-Jun as available) | EAnon NAnon NAm | Skill Score | Combined Snow Course + Gamma SWE |
| Snow Mass Trends (Sep-Jun) | NH midlatitudes NH arctic NHmon | Intercomparison | Snow mass trend evaluation ensemble |

**Notes:** PMW refers to SWE estimated from satellite-observations of passive microwave brightness temperatures.

The majority of tests use a 2-component skill score for evaluation. For each product its similarity to the specified reference

data is determined by two independent measures of "distance", one based on bias, the second based on a combined measure of product correlation and variance. Full details are provided in subsect. 2.4. The two distance measures are combined in quadrature into a total distance, and the distribution of total distances among all products is used to determine those which fall





into the upper 90$^{th}$ and lower 50$^{th}$ percentile of the distribution respectively. Any products performing above the 90$^{th}$ percentile are awarded +1 point; those performing below the 50$^{th}$ percentile are penalized -1 point.


For the trend evaluations, we employ an intercomparison approach relying in part on expert judgement. Seven products are chosen to form an evaluation ensemble for trends (described in subsect. 2.2.6) determined by the products that exhibit the most consistent seasonal evolution of snow mass trends across all three analyzed regions. These products are awarded + 1 point for each of the three regional tests. The remaining products can have minor, moderate, or substantial differences in the seasonality

or magnitude of their regional trends as discussed in Sect. 4. Those with minor differences in a particular region (generally within the spread of the evaluation ensemble) are also awarded +1 point. Products with substantial differences from the ensemble (outside the ensemble spread throughout the entire season) receive penalties of -1 point for that region. In cases where the differences are judged to be marginal the product is not awarded a point or penalized.

In Sect. 4 we rank all the products by their total points with discussion provided to indicate overall conclusions and performance idiosyncrasies for various products and product families.

## 2.3 Reference Data

### 2.3.1 Combined snow course and airborne gamma SWE datasets

We use a combined snow course and airborne gamma SWE reference data set recently cross validated in (Mortimer et al., submitted) and listed in the Data Availability Sect. The data are available over the 1979-2020 period with broad spatial coverage (Fig. 1) provided by the complementary availability of the two types of measurements. The confidence with which the reference data can be used to evaluate other SWE products differs somewhat according to terrain type. Mortimer et al. demonstrated that when evaluating gridded products across North American non-mountainous terrain with overlapping

reference data availability, both measurement types (snow course and airborne gamma SWE) yield product errors consistent with one another. In mountainous terrain the evaluated product errors differ according to the reference measurement type, primarily due to the respective range of SWE magnitudes each method tends to sample. Despite the differences in error magnitudes, choice of reference data type was shown to have little impact on *relative* product performance (i.e. product rankings). It is therefore possible to obtain robust relative performance measures across both mountainous and non-

mountainous terrain of North America. While airborne gamma measurements are only available over North America, snow course measurements are available across both the North American and Eurasian continents, however coverage across Eurasia mountain regions is inadequate.



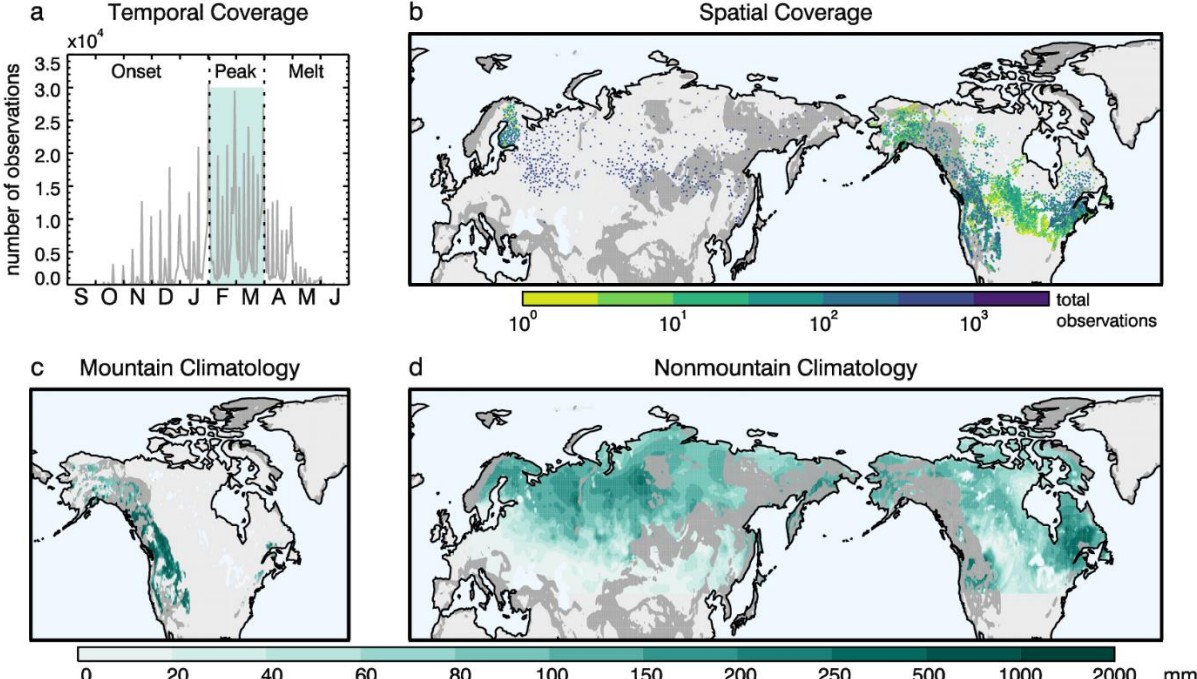

**Figure 1 a) Temporal coverage of combined snow course and gamma reference SWE measurements. b) Spatial coverage of combined snow course and gamma SWE (colors show total observations available at that location over the 1979-2020 period). c) March mountain climatology based on available combined snow and gamma SWE. d) March nonmountain climatology from bias-corrected GlobSnow version 3.**

Owing to these characteristics, we consider three distinct regions of the Northern Hemisphere to evaluate product variability: nonmountainous Eurasia (hereafter EAnon), non-mountainous North America (hereafter NAnon), and mountainous North America (hereafter NAm). While the Eurasian region does contain some data at mountain sites, the majority are situated in nonmountainous locations and the evaluation results will principally reflect those characteristics. Because the majority of data is available during Feb-Mar (Fig. 1a), we pool all data as available during these months into a single season. Pooled data available prior to February is considered a distinct 'onset' season, while pooled data available from April onwards is considered to belong to the 'melt' season. For a given season, this selection of pooled data results in a sequence of SWE values that combines aspects of spatial variability (the reference data locations are at specific locations across the region), interannual variability constrained by data availability (some years will be missing at given locations), as well as seasonal SWE evolution (when reference data is available at multiple times within the subseason of interest).

Evaluations of mountainous climatological SWE (limited to NA) also use the combined snow course and gamma SWE, however the data is first interpolated to a regular grid. We pool the subset of NA mountain reference data available during



March only (for consistency with the bias-corrected GlobSnow v3 data described below) and extrapolate it to a regular 0.5° x 0.5° output grid using an inverse distance method. The climatological SWE field determined by this process is shown in Fig.

1c. When comparing this reference climatological field to those from gridded products only reference locations with a non-zero SWE field value are used to calculate similarity; the remaining points are masked in each test product.

### 2.3.2 Bias-corrected GlobSnow v3 data

Bias-corrected GlobSnow v3 data (Luojus et al., 2021) is available between February and May and represents a spatially and

temporally continuous reference dataset currently available to assess climatological SWE across non-mountainous regions. The reference data is based on the monthly mean climatology of the GlobSnow v3 product that has been bias-corrected using a subset of the snow course data discussed in subsect. 2.3.1. Because the spatial coverage and sampling frequency of snow course data is optimal during March, we limit our analysis to that month (Fig. 1d). Because this reference data is itself gridded, it provides complete spatial coverage to assess product performance on climatological SWE in non-mountainous regions,

which would not be possible using the combined snow course and gamma data.

### 2.3.3 Evaluation ensemble for snow mass trends

We construct an ensemble using seven gridded reanalysis-type products selected for two criteria. First, we choose products forced by as broad a range of meteorological analysis fields as possible, and secondly, we choose products whose snow mass

trends are as consistent with one another as possible. It will be demonstrated in Sect. 4 that the reanalysis-type products which employ assimilation of surface snow information all have seasonal incongruities with one another. Therefore, the seven products chosen for this ensemble are all from model configurations without assimilation of surface snow observations or satellite information: ERA5-Land, Crocus-ERA5, BTIM-ERA5, BTIM-JRA55, MERRA, BTIM-ERA-Interim, and Crocus7. We exclude products forced by MERRA2 due to potential issues with the precipitation fields which are discussed in Sect. 4.

Because the forcing data is a stronger influence on the interannual variability and thereby on the seasonality of snow mass trends than the choice of snow model/scheme (Mudryk et al., 2015), we average SWE anomalies from the three products that use ERA5 forcings (ERA5-Land, Crocus-ERA5, BTIM-ERA5) into a single anomaly field and those from the two products that use ERA-Interim forcings (BTIM-ERA-Interim and Crocus7) into a second anomaly field. These together with the BTIM-JRA55 and MERRA anomalies produce four estimates of historical SWE anomalies distinguished by choice of forcing data.

We compute trends for each of these four anomaly fields and use both the 4-member mean trend field as well as spread in the regionally aggregated snow mass trends among the four members to determine consistency with snow mass trends from other gridded products in Sect. 4.



### 2.3.3 Independence of reference data and evaluated gridded products

While the majority of the gridded products evaluated here are completely independent from all the reference data discussed above, we discuss a few exceptions here. First, it is evident the standard GlobSnow v3 product is not independent of the bias-corrected version used to assess product climatologies across NHnon. Furthermore, given that the four GS/CCI products have a shared development history with strong similarities to one another, in the evaluation of NHnon climatological SWE, we do not rank these four products but only use them to guide interpretation of how well the remaining products perform. We also

point out that while the GS/CCI products as well as ERA5 and ERA5-Snow assimilate available weather station snow depths across both NH continents, these are in situ measurements and differ in both measurement frequency (sampled approximately daily versus once- or twice-monthly) and representative scale (being point measurements versus transects) from the snow course SWE measurements in the combined reference data. Therefore the aforementioned gridded products are explicitly independent of the reference data. Snow_cci v2 is an exception to this statement as in additional to in situ snow depth

measurements, it also incorporates extrapolated snow-course-derived snow density information (Venäläinen et al., 2021) within the SWE  retrievals. Thus it is not completely independent of the combined reference data set.

### 2.4 Skill scores and target diagrams

We use *skill target diagrams*, adapted from (Jolliff et al., 2009) in order to rank the similarity of the gridded products to the

reference data using a normalized distance measure, itself calculated from three independent statistics: the product bias $b$ (mean difference from the reference data), the product correlation with the reference data $R$, and the ratio of product standard deviation (sometimes referred to as the *amplitude*) to that of the reference data $\sigma_* = \sigma_x / \sigma_r$. Note that the latter two statistics are related to one another through the normalized unbiased root mean squared error $uRMSE_*$, as as

$$uRMSE_*^2 = 1 + \sigma_*^2 - 2\sigma_* R.$$

The above equation is the standard relationship used to relate $\sigma_*$ and $R$ on a Taylor diagram (Taylor, 2001) measuring the unbiased RMSE in units of the reference data standard deviation. Here, we employ skill target diagrams, which provide improved rankings compared to Taylor diagrams in two ways. First, they account for product errors in bias which are not represented on a Taylor diagram. Secondly, they use a skill score that more appropriately weights the pattern correlation and

amplitude compared to uRMSE, which otherwise preferentially ranks low-amplitude patterns above high-amplitude patterns given comparable correlations. We employ a modified version of the skill target diagram described in Jolliff et al.. The diagram is organized such that distance from the origin is a measure of the combined distance in two independent skill scores $S_{total} =$





$\sqrt{S^2_{pattern} + S^2_{bias}}$. Akin to the bullseye of a shooting target, the closer the squared distance of the independent error measures

are to zero, the lower the total error.

The first skill score combines the product's errors in amplitude and correlation as

$S_{pattern} = 1 - \frac{2(1+R)}{(\sigma_* + 1/\sigma_*)^2}$ .

This formulation is a standardly employed skill score ranging from 0 to 1 that can be used in place of uRMSE to better weight errors in amplitude and correlation (e.g. see Taylor et al. 2001). As in Jolliff et al., values approaching zero indicate superior skill (a reversal of the typical convention, used here so that the score measures distance from the origin). The second skill score measures the errors in bias as

$S_{bias} = \frac{|b|}{uRMSE_{max}}$ ,

where $uRMSE_{max}$ represents the maximum uRMSE among the ensemble of products evaluated (in absolute rather than normalized units). Our formulation differs from Jolliff et al. who use $S_{bias} = b/b_{max}$ where $b_{max}$ is the maximum bias in the product ensemble. We argue that scaling by $b_{max}$, can overweight the contribution of bias to the total skill distance, $S_{total}$,

whereas normalizing by a measure of the ensemble uRMSE accounts for the proportion of the total RMSE contributed by the bias since $b/uRMSE_{max} = (b/b_{max})(b_{max}/uRMSE_{max})$ and $RMSE^2 = uRMSE^2 + b^2$ . If $b_{max} \sim uRMSE_{max}$ , then $S_{pattern}$ and $S_{bias}$ will contribute equally to $S_{total}$ since there will be an ensemble member for which $S_{bias} \sim 1$. However, if $b_{max} \ll uRMSE_{max}$, the total skill distance should be determined principally by $S_{pattern}$, which is the case as formulated here, but not as formulated in Jolliff et al..


Computing the combined skill distance described above requires three input statistics: bias, correlation, and standard deviation. These were calculated for each product as follows. Product climatologies, which are evaluated using gridded reference data (Sect. 2.3.1, Figs. 1c and d), were calculated on their native grids and subsequently interpolated to a 0.5° x 0.5° output grid to match up with the reference data. For the tests of product variability, we performed a station-wise comparison. The reference

data was matched up in time and space as specified in Mortimer et al. 2023 (but with snow courses and gamma data pooled together) over the full 1980-2020 period. To avoid oversampling in regions with dense coverage, for a specified day of interest we average reference sites within a single product grid cell and then to 100km spacing. A sample location is determined to be mountainous or nonmountainous according to a combination of the Global Mountain Biodiversity Assessment (GMBA) Mountain Inventory v2 (Snethlage et al., 2022) with a 25km buffer applied and a 2° slope mask derived from the GETASSE30



DEM. For both gridded and station-wise reference data the above procedures result in a sequence of N paired SWE samples (reference data samples denoted $r_i$, product data samples denoted $x_i$) from which we calculate:

$$b = \frac{1}{N}\sum_i x_i - r_i$$

$$\sigma_x{}^2 = \frac{1}{N}\sum_i (x_i - \bar{x})^2$$

$$\sigma_r{}^2 = \frac{1}{N}\sum_i (r_i - \bar{r})^2$$

$$R = \frac{\sum_i (x_i - \bar{x})(r_i - \bar{r})}{\sqrt{\sum_i (x_i - \bar{x})^2 \sum_i (r_i - \bar{r})^2}}$$

$$RMSE^2 = \frac{1}{N}\sum_i (x_i - r_i)^2.$$

## 3 Results

### 3.1 Climatological SWE evaluations

Before presenting the performance of individual gridded products on the series of tests described in Sect. 2.2, we first illustrate how the spread in climatological snow mass across both mountainous and nonmountainous regions of the NH varies among the products by sorting the products into four groups. The first group we consider are five previous generation reanalysis-derived products (now deprecated): ERA-interim, B-TIM-ERAint, Crocus7, Crocus8, and MERRA (denoted "Reanalysis Group 1" in Fig. 2). For comparison in the second group ("Reanalysis Group 2") we consider gridded SWE products based on
the current generation of reanalyses: ERA5, ERA5-Snow, ERA5-Land, Crocus-ERA5, MERRA2, B-TIM-ERA5, B-TIM-MERRA2, B-TIM-JRA55. The third group contains the GS/CCI (EO) products and the JAXA EO product (shown separately in Fig 2). The four GLDAS products are also plotted individually as they have large biases as illustrated in the figure and also as analyzed in the subsequent tests below.

Figure 2 illustrates that snow mass across nonmountain regions has, on average, increased in the current generation of reanalysis-based products from the versions analyzed in (Mudryk et al., 2015). The updated products agree better both with one another and with nonmountainous snow mass aggregated from the bias-corrected GlobSnow version 3 SWE reference




data (subsect. 2.3.2). Snow mass estimated from non-bias-corrected GS/CCI products have lower snow mass on average during March than the current generation of reanalysis-derived products. Across mountain regions, the spread and mean values have

increased among the newer reanalysis-type products. These increases are due to deeper SWE conditions in the Crocus-ERA5 and ERA5-Land products specifically, whereas the remaining Group 2 products have a similar range of snow mass estimates as the Group 1 products (not shown). JAXA is the only Earth-Observation product that attempts to estimate SWE in mountain regions, but estimates unrealistically low snow mass compared to that that found in any of the reanalysis-type products other than the GLDAS products. Figure 2 also illustrates climatological snow mass from the four GLDAS products. GLDAS v2.0

output from either land model (Noah or CLSM) has unreasonably low snow mass across both nonmountainous and mountainous regions. Even if data assimilation is used as for the GLDAS v2.2 output using CLSM the nonmountainous snow mass remains unreasonably low. However GLDAS v2.1 using Noah (Fig 2, dark green cross), which replaces the Princeton precipitation forcing used for all other versions with the gauge-based GPCP v1.3 precipitation product, has snow mass that is much more consistent with the other products.


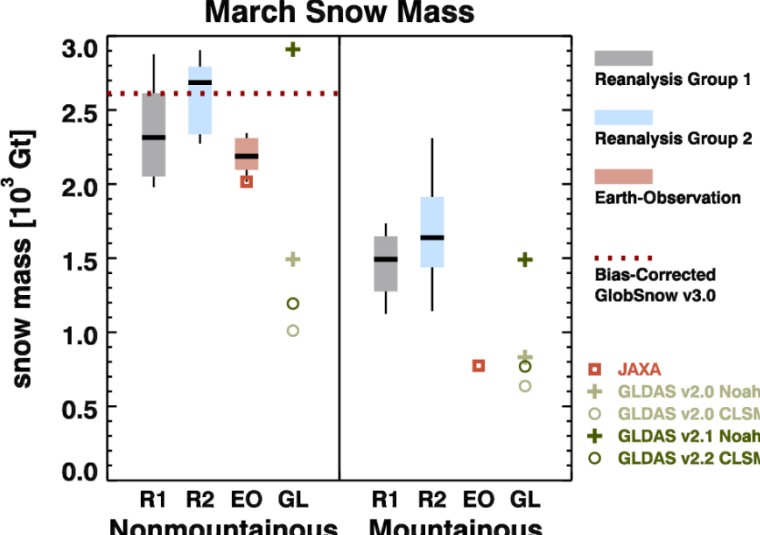

**Figure 2 Spread in nonmountainous and mountainous March snow mass for various groupings of products. JAXA and GLDAS products are considered separately as denoted by symbols.**

In Fig. 3 we examine the relative ability of products to capture the correct spatial distribution of climatological SWE across both nonmountainous and mountainous terrain. Products are evaluated using skill target diagrams (after Jolliff et al., 2009; see Section 2 for details) with Taylor diagrams also shown for reference. Figure 3 illustrates that when assessed using a Taylor diagram roughly half the products have minimal spread in their skill at reproducing the correct spatial distribution of climatological SWE in nonmountainous regions and perform nearly as well as the GS/CCI products (red squares), which are





shown on the plot but are not ranked due to their similarity to the bias corrected GlobSnow version 3 reference data (see

subsect. 2.3.4). More discernment among the products is apparent on the target diagram, which illustrates that ERA-Interim-

Land, JAXA, JRA55 and three of the four GLDAS products are in the lower half of the product distribution and that among

the remaining products there is a range of positive and negative biases. Note that using the total skill distance (target diagram)

yields different rankings from using uRMSE errors (Taylor diagram). This difference is especially important in mountainous

regions where the products' ability to capture the variance in the climatological SWE distribution varies dramatically. As

highlighted in Sect. 2, the fact that essentially all products underestimate the spatial variability in climatological SWE

compared to the reference data affects the uRMSE-based rankings. In particular, despite having both modestly improved

correlation and substantially improved spatial variability compared to the reference data, both Crocus-ERA5 and ERA5-Land

have higher uRMSE values in mountainous regions than many of the other products (Fig 3, upper right). When ranked by their

total skill distances instead (Fig 3, lower right) these are the two best performing products in mountain regions performing

above the 90th percentile among the range of products. We also note that mean bias forms a much larger fraction of the total

mean error in mountainous regions compared to nonmountainous regions (up to roughly 50% of the uRMSE versus 5% in

nonmountainous regions). For these reasons we use only the skill target diagrams in the subsequent analysis and the combined

skill score to rank the products.


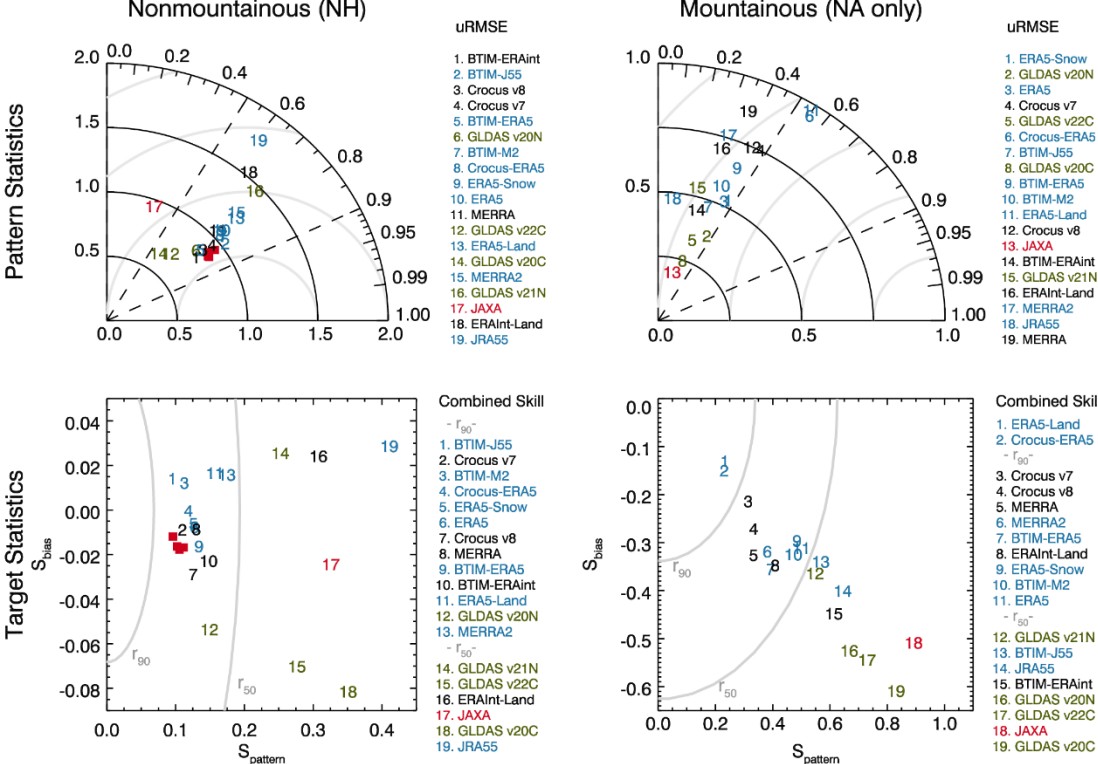



**Figure 3 Taylor plots (top) illustrate performance ranked by uRMSE (distance from reference data measured in units of the standard deviation and shown by the concentric grey circles) in nonmountainous (left) and mountainous (right) regions. Target plots (bottom) illustrate performance ranked by total skill distance (skill scores of zero represent no difference from the reference data in terms of pattern statistics or mean bias). Grey curves indicate the 90th and 50th percentiles. Red squares denote the performance of the GS/CCI products which are considered "close" to the reference data in nonmountainous regions. Colors reflect the groupings from Fig. 1.**

### 3.2 Time-varying SWE evaluations

The next series of tests evaluates the gridded products on their ability to capture time-varying SWE during three portions of the seasonal cycle. We initially examine performance near the seasonal peak (Feb-Mar). Before presenting the overall skill rankings for this evaluation we first examine separate rankings of uRMSE, correlation, and bias to provide a sense of how they relate to one another. Figure 4 illustrates performance across nonmountainous terrain in North America compared to nonmountainous terrain in Eurasia. In general, products have poorer performance over North America than over Eurasia. This may occur since the range of reference SWE sampled is higher in North America and this is a strong control on product bias and RMSE (see Mortimer et al., submitted). Product performance evaluated by either uRMSE or correlation are similar to one another: product rankings 1-7, 8-15, 16-20, and 21-23, respectively, all contain the same subsets of products when evaluated using uRMSE as with correlation. In contrast, bias is a poor discriminant of product performance in nonmountainous terrain. Products may have low bias but high uRMSE and low correlation due to poor representation of SWE anomalies (JAXA, JRA55).

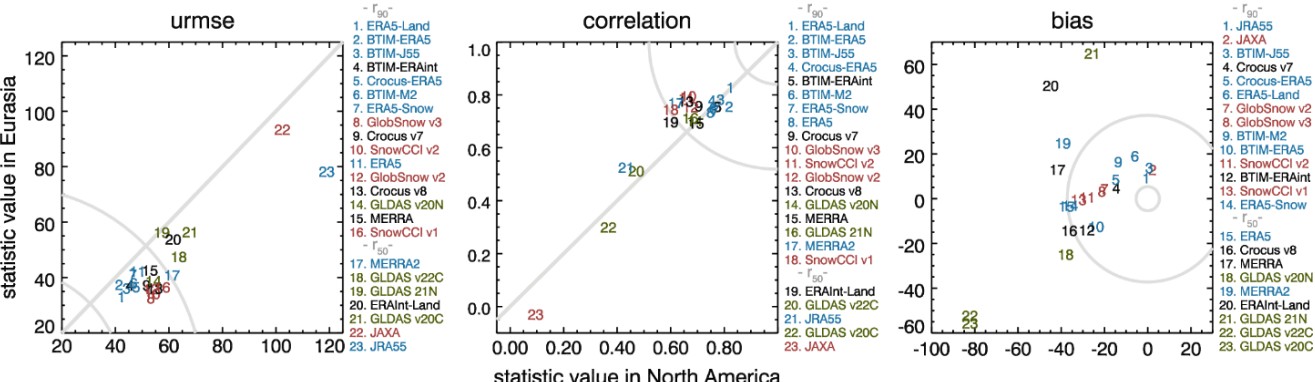

**Figure 4 Product-wise performance for peak SWE in North America versus Eurasia evaluated over nonmountainous regions. Products are ranked based on their North American and Eurasian statistics added in quadrature. Grey curves denote the 90th and 50th percentiles of the product distributions; these two percentiles are listed among the ranked products where they occur.**



For this reason, in Fig. 5 we employ the same target plots as presented for climatological snow mass and which account for combined errors of bias, uRMSE, and correlation. Consistent with Fig. 4 and Fig S1, the latter of which shows results for uRMSE, bias, and correlation metrics over mountainous terrain, the combined skill distance in Fig. 5 illustrates that product performance is generally best over nonmountainous Eurasia, worse over North American nonmountainous terrain, and worse again over North American mountainous terrain. Across Eurasia no product substantially outperforms another (none are above the 90th percentile), although most of the worst performing products also fall below the 50th percentile across all three combinations of continent and terrain (JAXA, JRA55, and three of the four GLDAS product versions). ERA5-Land and Crocus-ERA5 display the greatest skill in North American mountainous terrain and have good to excellent performance in nonmountainous regions of Eurasian and North America as well. While the BTIM suite of products are typically top performers in nonmountainous North America, they perform more modestly across North American mountainous regions. The GS/CCI products have good performance across Eurasia, but their performance is poorer across North America. As seen for climatological SWE (Fig 3), in mountainous terrain product bias *is* associated with overall performance versus in nonmountainous terrain (see also Fig. S1).

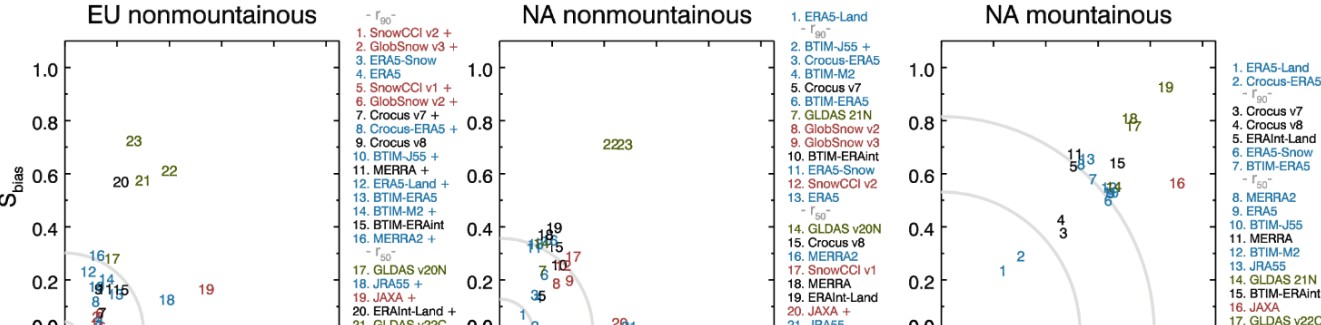

**Figure 5 Target plots based on statistics for peak SWE from temporally and spatially varying data for available continents and regions. Products are ranked based on total skill distance (skill scores of zero represent no difference from the reference data in terms of pattern statistics or mean bias). Grey curves denote the 90th and 50th percentiles of the product distributions; these two percentiles are listed among the ranked products where they occur.**

In Fig. 6, we examine if the product-wise performance analyzed in Figs. 4 and 5 near seasonal peak SWE (Feb-Mar) remains consistent during the onset and melt seasons. Figure 6 illustrates that the product accuracy tends to worsen as the snow season progresses: on average both the bias and pattern skill decrease corresponding to increasing uRMSE, decreasing correlation, and increasing magnitude of bias. However, the products that have better performance when evaluated near seasonal peak SWE (when the most reference data is available thereby yielding more accurate statistics) tend to have better performance



during the onset and melt seasons. In particular, the pattern skill component assessed during peak season is also a reasonable indicator of performance during both onset and melt. In contrast the evolution of seasonal bias can change substantially among
the products in especially in nonmountainous regions.

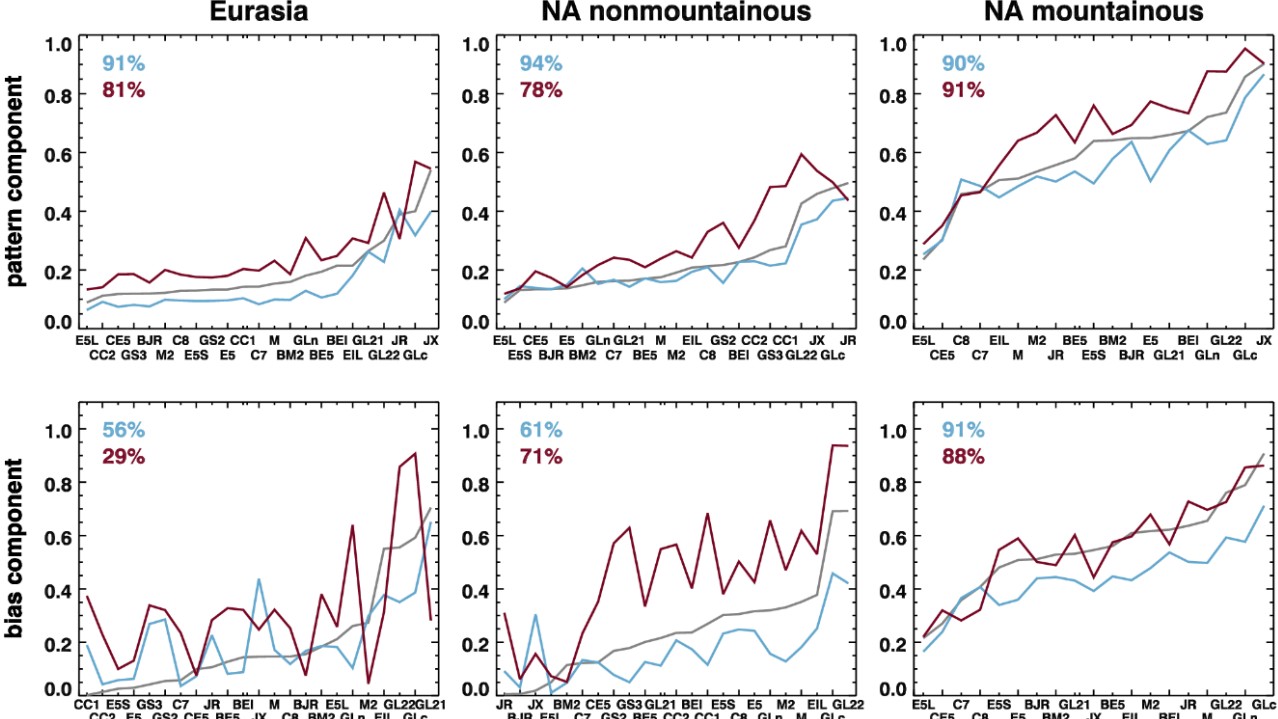

**Figure 6 Seasonal evolution of skill components by continent and region. Products ranked by FM performance (grey; x-axis labels) with corresponding performance shown during onset (blue, SONDJ as available) and melt (red, AMJ as**
**available) seasons. Numbers displayed in corners show percentage of onset and melt performance explained by FM performance.**

**3.3 Trend Evaluations**

Finally, we evaluate differences in product trends using a quantitative intercomparison approach that also incorporates
elements of expert judgment. As detailed in subsect. 2.3.3, we select the gridded products which have the most consistent seasonal evolution (magnitude and timing) of aggregated snow mass trends across three mutually exclusive regions: nonmountainous midlatitudes (south of 60N), nonmountainous Arctic (north of 60N), and NH mountainous regions (all latitudes). The trends of these products are averaged according to their forcing meteorology and together they form an "evaluation ensemble" with spread as shown in Fig. 7: MERRA (forced by MERRA meteorology), BTIM-JRA (forced by



JRA55 meteorology), BTIM-EIL and Crocus7 (forced by ERA-Interim meteorology), and ERA5-Land, BTIM-ERA5, and
Crocus-ERA5 (forced by ERA5 meteorology).

The remaining gridded products are compared in each of the three regions with the evaluation ensemble and assigned scores
as described in Sect. 2.2. Figure 7 (top row) first compares trends from JRA55 and BTIM-JRA55. The two products use the
same forcing meteorology but differ in their snow schemes and whether or not they assimilate passive microwave-derived
snow extent information (JRA55 does but BTIM-JRA55 does not). We argue it is unlikely that the differences in trend
magnitudes and timing shown are due to differences in the snow scheme employed, nor can they be due to differences in
meteorological forcing. This suggests that the assimilation of passive microwave data employed by JRA55 is causing
anomalous trends in alpine and midlatitude regions. JRA55 trends in Arctic regions still have anomalous signals in their
seasonal evolution, however we assess the agreement in that region as marginal.

Next we examine the suite of products related to ERA5 (second row of Fig. 7). Because they don't assimilate land surface
information, we are reasonably confident that ERA5-Land, Crocus-ERA5, and BTIM-ERA5 do not exhibit spurious seasonal
timing and their average is one of the components in the evaluation ensemble. The standard ERA5 reanalysis output, by contrast
is known to contain spurious trends due to its assimilation of IMS data which occurs from 2004 onwards (Mortimer et al.,
2020; Ochi et al., 2023), resulting in overly negative trends across all three regions. ERA5-Snow is an "offline" product which
was forced by ERA5 analysis fields in an uncoupled configuration. It was produced to allow for assimilation of weather station
snow depth information but to avoid the abrupt incorporation of IMS information from 2004 onwards. While ERA5-Snow
trends have better agreement with the evaluation ensemble than ERA5, they are still more strongly negative over Arctic
regions. We assess this level of disagreement as marginal, in comparison to that shown by ERA5 trends in all three regions.

The third row of Fig. 7 compares snow mass trends from the original MERRA reanalysis output with those from the updated
MERRA2 product and the BTIM snow scheme forced by MERRA2 temperature and snowfall. The latter two trends have
similar timing and magnitudes to one another, but across midlatitude and mountainous regions their magnitude is much weaker
than those from the evaluation ensemble. The fact that BTIM-MERRA2, which is driven by the same temperature and
precipitation forcing as MERRA2, has similar snow mass trends to those from the reanalysis suggests that the temperature or
precipitation forcing or both are inconsistent with the other products in the evaluation ensemble (two generations of ECMWF
analysis, the JRA analysis, as well as the previous generation of MERRA analysis).








**Figure 7 Evaluation of snow mass trends by region (columns) for families of products (rows). Evaluation ensemble shows spread among MERRA, BTIM-EIL, Crocus7, BTIM-J55, ERA5-Land, BTIM-ERA5, and Crocus-ERA5 combined to equally weight meteorological forcings as described in Sect. 2. For each row, trends are calculated over the period denoted on the left, chosen based on the period that the plotted products are available; therefore the grey shading denoting the evaluation ensemble spread differs somewhat among the rows. Numbers denote total (column 1) and regional (columns 2-4) trend scores based on arguments presented in the text.**






The fourth row of Fig. 7 compares trends from two separate groupings, the two GLDAS 2.0 products, and the three ERA-Interim forced products: ERA-Interim-Land, Crocus7, and BTIM-ERA-Interim. The ERA-Interim products are consistent over midlatitude regions but ERA-Interim-Land has inconsistencies over Arctic and mountainous regions where its trends are weaker than those of the evaluation ensemble. The two GLDAS products are marginally consistent with the evaluation ensemble over mountain regions but have overly weak trends in mid-latitude and Arctic regions.

Finally the bottom row of Fig. 7 compares the evaluation ensemble with trends from GS-CCI products, which are the only Earth-Observation based products in our product suite that provide long enough historical records to calculate trends. Because the GS/CCI products do not provide SWE estimates in mountainous regions, we use mean anomalies from the evaluation ensemble in those regions to determine total NH trends, which allows us to observe how differences in the midlatitudes and Arctic regions combine hemispherically. The bottom row of Fig. 7 indicates that there are differences between the evaluation ensemble and the GS/CCI products during Mar-May in both the midlatitudes and the Arctic, but that they tend to offset one another when combined in the total NH mass trends. Three of the four GS/CCI products also show stronger midlatitude trends during snow onset in November and December excepting the most recent version (Snow_cci v2). In Fig. S3, we connect this difference across the midlatitude region to temporal discontinuities in the early and late parts of the record that have been improved but not eliminated in the most recent product. The differences in Arctic trends during Mar-May are likely related to reduced availability of in situ data during this time of the year combined with reduced ability of the satellite algorithms to retrieve SWE once the snowpack begins to melt (Mortimer et al., 2022). Despite these suspected issues, we consider the trends to be generally in agreement with the evaluation ensemble across the Arctic. Over midlatitudes we consider the trends to be in disagreement (see Fig. S3 for further evidence) but assign marginal disagreement to the Snow_cci v2 product to acknowledge its improved performance compared to previous versions. We also note that improvements to the snow masking (Zschenderlein et al., 2023) feeding into successor versions of Snow_cci (e.g. the forthcoming version 3 SWE product) better reproduce the snow mass trends seen in the evaluation ensemble across midlatitudes and Arctic regions during snow onset (November to January; K. Luojus, private comm.)

## 4 Overall Performance and Discussion

Figure 8 shows the complete list of hemispheric products organized by overall performance. The overall product rank is determined by a product's cumulative score on all tests divided by the number of tests on which it was evaluated. This allows the assessment to be agnostic about products whose performance in a particular test was unable to be evaluated. For example, JAXA, GLDAS v2.1, and GLDAS v2.2 did not have enough available years of data to calculate trends while the GS/CCI products are not available across mountainous regions and so are untested there). For comparison, we also provide a second set of rankings that only reflects the tests based on skill scores (the trend intercomparison assessment is excluded). The products





with the best and worst performance are ranked similarly in these two sets of rankings, however the positions of products with

average performance (ranks 4-16) is influenced by the trend intercomparison. While we believe the trend intercomparison

provides additional information by which these products can be compared, we leave it to readers to determine for themselves

whether they wish to include this additional information.

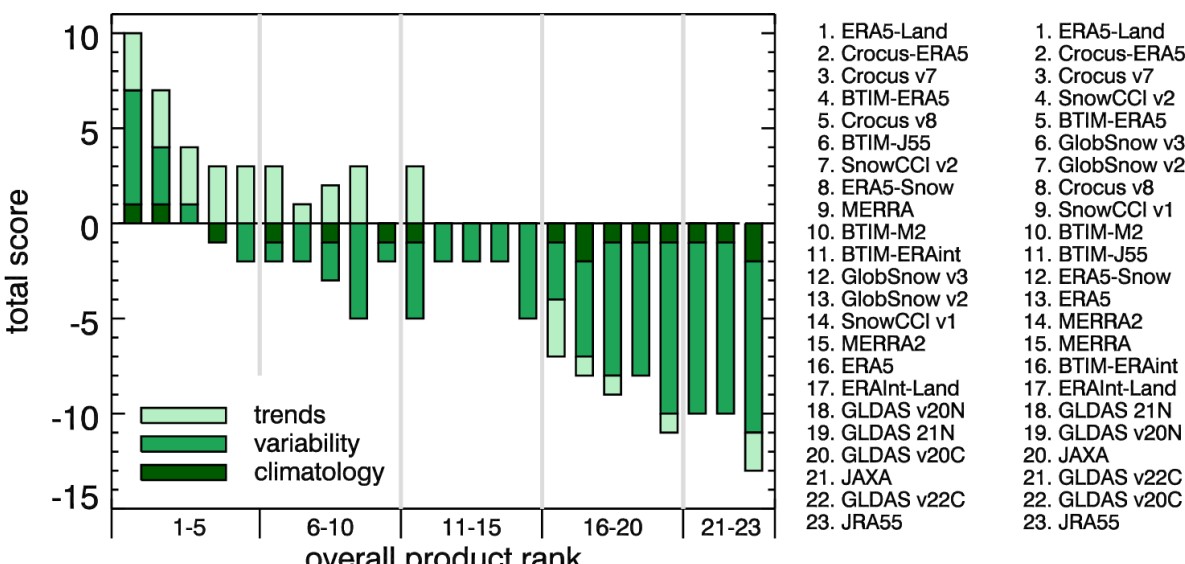

**Figure 8 Ranked overall performance based on all tests (x-axis) broken down by category: climatology (dark), variability (medium) and trends (light). The first ranked list is based on all test categories; the second ranked list excludes the trend evaluation.**

The top performing SWE products are ERA5-Land followed by two versions of the Crocus model (versions using forcing data

from both the previous ECMWF ERA-Interim reanalysis and the updated ERA5 reanalysis). These products benefit from a

comparatively high horizontal resolution in the case of ERA-Land (10km) or by a high vertical resolution in the case of the

Crocus snow model (approximately 10 snow layers in the global product, regional implementations may use up to 30 layers).

This may be a reason for their strong performance especially in the highly variable SWE of the North American mountain

regions. These products also benefit from the absence of surface snow assimilation which negatively impacts snow mass trends

of other products.

The B-TIM suite of products, which are based on a simple temperature index scheme, generally have good performance in

nonmountainous regions, where they are consistently in the top half of the rankings, indicating that these simple products have



value across non-mountain areas (Figs. 3 and 5). Furthermore, the trend intercomparison (Fig. 7) suggests that they are also a

valuable tool for detecting anomalous SWE trends in other products, at least on a regionally aggregated basis.

All four GLDAS products perform very poorly hemispherically (Fig. 5) due in part to large biases (Figs. 2 and 4), although GLDAS v2.1, which uses different precipitation forcing than the other three versions, performs better over the continental United States (Fig. S2). While it is tempting to extrapolate regional performance, this product provide a good counter-example

where doing is particularly detrimental and we do not recommend their use, particularly outside of the continental United States.

The GS/CCI products have better performance over Eurasia than North America (Fig. 5). This may result because nonmountainous reference data over North American spans a larger range of SWE magnitudes. Evaluating the GS/CCI

products using larger SWE magnitudes will decrease their performance since their algorithms' SWE retrievals tend to saturate above 150mm in comparison to many of the other gridded products (Dong et al., 2005; Mortimer et al., submitted). The influence of these effects would be less prominent over Eurasia where the reference SWE magnitudes are lower. It is also possible that compared to North America, Eurasia may have more commonality in how the locations and overall coverage of the reference data aligns with in situ snow depth measurements assimilated by the products. The latter is essential information

for the GS/CCI algorithms to perform accurately. If locations of reference data across North American tend to be further from locations with assimilated data compared to Eurasia, this would also lower product performance. Because of these considerations we suggest the evaluated accuracy of GS/CCI products over North America is more reflective of their true performance.

Finally, the trend analysis indicates that for the ensemble of products evaluated here, all attempts to assimilate snow information from surface and/or satellite measurements lead to a deleterious influence on snow mass trends (e.g. ERA5 and JRA-55). The influence of the assimilation techniques employed on snow mass trends is not minor or localized but leaves clear signals even in the trends of regionally or hemispherically aggregated snow mass. While assimilation of surface information may improve instantaneous, local measures of the overall performance of a reanalysis system, it reinforces that reanalysis

centers should provide multiple product streams: not only those that provide the best instantaneous estimates as needed for prediction applications but also temporally consistent historical estimates, which are needed for climate applications. In some ways, the series of GLDAS products provides a good model for this sort of treatment, with an open loop suite of output without assimilation and another assimilated product. Unfortunately, at present the forcing data used for the multiple GLDAS product streams differ and there is insufficient overlap of the analysis periods to permit attribution of differences in trends between the

products to the presence or absence of data assimilation.





The relative overall rankings shown in Fig. 8 are meant to function as a guideline only. The evaluations used to determine the rankings are based on current state-of-the-art reference data (snow transects/courses and airborne gamma measurements) that are reasonably representative of the spatial scale of gridded products. The range of evaluations have separately considered several large-scale aspects of Northern Hemisphere terrain, geographical coverage, and seasonal evolution. Nonetheless, the fact that the reference data has imperfect spatial and temporal coverage may influence the evaluation of some products. In particular we stress that product performance across Eurasian mountain regions has not been evaluated and any idiosyncrasies in product performance across those regions would alter the rankings as presented.

We also acknowledge that for some tests, there was not a sharp boundary in performance among groups of products. Hence the dividing line between the top and bottom half of the rankings is blurred. That said, the number and breadth of tests presented should help ensure that our conclusions of which products are superior performers are robust.

**Conclusions**

An expanded reference dataset (Fig. 1), consisting of snow course and airborne gamma measurements (Mortimer et al., submitted), combined with a novel evaluation strategy allowed for a comprehensive assessment of 23 gridded SWE products. We adapted skill target diagrams (Jolliff et al. 2009) to rank products according to their ability to represent SWE climatology (Fig. 3), variability (Fig. 5), and trends (Fig. 7). Most products evaluated can reasonably represent the climatology and variability of nonmountainous SWE but have substantially lower skill in mountain regions (Figs. 3 and 5). The relatively poorer performance in mountain regions is consistent with previous studies (Fang et al., 2022; Kim et al., 2021; Liu et al., 2022; Snauffer et al., 2016; Terzago et al., 2017; Wrzesien et al., 2019) and points to a need for targeted mountain SWE products. For the ensemble of products evaluated, the assimilation of snow surface and/or satellite measurements has a deleterious influence on regional snow mass trends (Fig. 7). This result illustrates that products that accurately represent SWE climatology and variability may not be appropriate for trend analysis and vice versa, and reinforces that user needs and objectives must guide product selection.

**Data Availability**

Combined reference data available is available at https://zenodo.org/[currently being finalized; to be available at time of publication]. The bias-corrected GlobSnow version 3 product is available from https://www.globsnow.info/swe/archive_v3.0/. Gridded product SWE available as in Table 2.

| Product Name | Availability/DOI |
| --- | --- |
| B-TIM-ERA5 | DOI to be available at time of publication |
| B-TIM-JRA55 | DOI to be available at time of publication |





| | |
|---|---|
| B-TIM-MERRA2 | DOI to be available at time of publication |
| B-TIM-ERAint | From authors on request |
| Crocus-ERA5 | DOI to be available at time of publication |
| Crocus v8 | From authors on request. |
| Crocus v7 | From authors on request. |
| ERA5 | 10.24381/cds.adbb2d47 |
| ERA5-Snow | Available on request from patricia.rosnay@ecmwf.int |
| ERA5-Land | 10.24381/cds.e2161bac |
| ERA-Interim-Land | Deprecated. Author archival copy. |
| GLDAS v2.2 CLSM | 10.5067/TXBMLX370XX8 |
| GLDAS v2.1 Noah | 10.5067/E7TYRXPJKWOQ |
| GLDAS v2.0 CLSM | 10.5067/LYHA9088MFWQ |
| GLDAS v2.0 Noah | 10.5067/342OHQM9AK6Q |
| JRA-55 | https://jra.kishou.go.jp/ |
| MERRA2 | 10.5067/RKPHT8KC1Y1T |
| MERRA | 10.5067/YL8Z7MICQZF9 |
| Snow_CCI v2 | 10.5285/4647cc9ad3c044439d6c643208d3c494 |
| Snow_CCI v1 | 10.5285/fa20aaa2060e40cabf5fedce7a9716d0 |
| GlobSnow v3 | 10.1594/PANGAEA.911944 |
| GlobSnow v2.1 | https://www.globsnow.info/swe/ |
| JAXA-AMSR2 | Preliminary version provided as part of SnowPEx+. |
| | Available on request from rejkelly@uwaterloo.ca |


**Author Contributions**

LM and CM developed the general evaluation strategy, code to calculate statistics and performed the analysis. LM, CD, PK, and AEC developed the trend intercomparison strategy. LM prepared the manuscript with contributions from all co-authors.

**Competing Interests**

Some authors are members of the editorial board of journal The Cryosphere.



## Acknowledgements

This work was initiated through the ESA-funded SnowPEx+ project.

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
