# Peer review of "Benchmarking of SWE products based on outcomes of the SnowPEx+ Intercomparison Project"

_EGUsphere, 2023_

## Referee Comment (RC1)

**Manuscript Title:** Benchmarking of SWE products based on outcomes of the SnowPEx+ Intercomparison Project

**General review:**

Mudryk et al. aim to evaluate 23 different SWE products based on how well they represent SWE climatology, variability, and trends across mountainous and non-mountainous regions in North America and Eurasia. Using existing and newly created reference datasets, the gridded products are scored using skill target diagrams, resulting in a series of Taylor and target plots, eventually leading to an average ranking of the SWE products. The methodology and technical approach to this evaluation is clear. However, the presentation of the datasets, methodology, and results is notably convoluted or disorganized at times, dampening the impact that this thorough analysis could have. In particular, a clear workflow figure could aid in the introduction of the overall evaluation strategy, where related text often references several other sections, causing for much back-and-forth within the manuscript. While repetition in stating the methodology is appreciated, sometimes the methods, results and associated discussion appear in a single section, making the information challenging to process. Thus, I include no major analysis comments and suggest the authors primarily focus on restructuring the manuscript for a clearer portrayal of meaningful results. My more pointed comments below should help with these review items.

**Detailed/line-by-line comments:**

Suggest writing out each abbreviation first (e.g., ILAMB, AMBER, IMS)

Table 1: Suggest placing a reference column and stating, prior to the table, that each "family" of SWE product will be discussed in more detail following the table and/or placing the text prior to the table.

Section 2.2

This section is generally challenging to follow, yet it is intended to set up the pertinent evaluation strategy. There is reference to various sections ahead of the current (e.g., full details referenced in section 2.4, reference to 7 products chosen for an ensemble). Perhaps a schematic of the workflow/evaluation scheme would be helpful. As such, it is also unclear when a point system was introduced (line 153-154).

Line 156: Unclear what exactly expert judgement is considered in this case

Table 2: Suggest including some justification as to how regions were selected. Unclear at what spatial scale these variables are evaluated.

Line 172: It would benefit the readership if the text stated the spatial extent in addition to the figure.

Figure 2 c-d: It is unclear what the scale bar is referencing, as it pertains to climatology. If this is "SWE climatology," or peak SWE magnitude, labeling it and stating so in the figure caption, as opposed to "climatology," would be helpful.

Line 219: It is unclear what "broad a range of meteorological analysis fields as possible" looks like for the 7 selected products, which are spelled out later in this paragraph. Could an example be provided compared to a product that was not selected?

Line 221: Consider removing the following sentence, as it again jumps ahead to several sections from the current and causes confusion: "It will be demonstrated in Sect. 4 that the reanalysis-type products which employ assimilation of surface snow information all have seasonal incongruities with one another." Suggest revising line 124 for similar reasons.

Section 2.4: Please number and format equations similarly, as there are many and some build off of one another. Reference to a figure here or in one of the citations may bring additional intuition to this methodology here, prior to seeing the results.

Line 249: Suggest a ½ to full sentence on why this approach was taken to rank similarity across products (were there other approaches in consideration?).

Line 353: Additional annotations on this figure would be helpful. For example, placing notation near Crocus-ERA5 and ERA5-Land on the upper right panel would aid in the necessary scanning between text and figure (especially since the numbers/rankings obviously change between panels). This comment extends to Figure 4 and 5. Suggest also reiterating what is represented on each axis in each figure and/or across panels, particularly in the Taylor plots and in reference to pattern statistics.

Section 3.3: Many of these beginning sentences/paragraphs, aside from the sentences explicitly referring to Figure 7, read as though they belong in the methods or discussion sections, which decreases the impact of the following results. There is a lot of information to unpack in Figure 7. The paragraph structure for each row is appreciated. Perhaps the authors would consider beginning each paragraph with the intended take home point, particularly for the bottom row of results.

Section 4: The final result, presented quite clearly, does seem as though it could live in the results section. Are there comparable results specific to the select regions (NA and Eu mountain and non-mountain)? This question also pertains to my next comment.

Line 535: Can the authors expand and be more explicit about "The relative overall rankings shown in Fig. 8 are meant to function as a guideline only"? The following conclusion also states that "user needs and objectives must guide product selection," however a lot of technical and thorough work went into the culminating Figure 8. Are there thus product recommendations and takeaways for users broadly and by region?

Discussion: There lacks a discussion on limitations to this assessment and consideration of other gridded SWE products.

---

## Author Comment (AC1)

Response to Reviewer 2
Responses in blue.

Thank you for the time and effort to review our study and the generally supportive conclusions. We have addressed your criticisms and suggestions below.

We also note the following 3 analysis changes that affect some of the product-specific statistics, but don't alter generalized rankings or conclusions about product performance:
1. There was an error in the skill score components calculated for Figure 3 that has been fixed (the bias and amplitude values used in the figure were incorrectly normalized, but the relative rankings were still correct).
2. We have altered the way we calculated skill scores for climatological SWE in the NAm region to be more similar to the time-varying statistics and to ensure the statistics reflect the native resolution of the individual products.
3. We added an overall scale factor to both S_bias and S_pattern that allows readers to assess how product performance varies across differing regions/terrain throughout the paper (essentially skill scores for a regional/terrain-specific test are scaled by the uRMSE of the worst performing product across all tests). Since this doesn't alter relative rankings on a particular test it only affects the perceived performance on the given test relative to the other tests. Text in the revised manuscript describes this.

Many of the SWE data applied in this study are based on snow accumulation/melt algorithms embedded in different reanalysis models. The derived SWE is therefore a result of the forcing data, primarily temperature and precipitation. When comparing SWE data from these different sources, I miss a discussion on the performance (evaluation scores) of the forcing data used for the various approaches since the SWE estimates will inherit some of their characteristics.
-This would be an interesting analysis but one we are unable to explore fully in this paper. Such an analysis would add quite a bit to an already complex paper and it can only help explain the performance of reanalysis data sets, not the Earth-observation products such as SnowCCI, GlobSnow, and JAXA. Even for the reanalysis datasets, it is clear it will not be able to explain some of the major elements of the performance. For example, ERA5, ERA5-Snow, BTIM-ERA5, and Crocus-ERA5 all use the same meteorology, but span a range of final rankings from 2 through 16 out of 23.

Another issue I feel is almost neglected in the discussion is the role of the native resolution of the gridded datasets. Snow is a property that shows large spatial and temporal variability. Even though the comparison is performed on a joint 0.5°x0.5° grid, the original resolution should have an impact on the estimates. The native resolution of the data sets should be added in table 1.

We answer this question in two steps. First, we note that only the climatological tests in the original analysis used regridded data. For the time-varying results (9 of the 14 tests), the

gridded products were analyzed at their native resolution (the sequence of reference-product pairs used to calculate bias, correlation and standard deviation are selected based on proximity to the reference data using the native resolution of each gridded product). The text in Section 2.4 has been revised to explain this more clearly. Based on this comment and to help simplify our methods we have also altered the NAm climatological test to be more similar to the regional SWE variability tests (and to therefore use the native resolution of each gridded product climatology). Secondly as illustrated in the figure below, we note that resolution is a surprisingly poor explanatory variable for predicting product performance. This is true even after removing products with spurious trends (diagnosed in our manuscript Fig 7 and shown in the figure below in a lighter color).

[Figure]

*Figure:* Relative performance of products according to total number of horizontal grid cells in the NH (log scale). Dark markers denote products with no diagnosed trend issues/discontinuities; pale markers denote products with diagnosed issues. Even removing products with diagnosed trend issues, and removing the remaining two GLDAS products as outliers, the connection to resolution is not strong with multiple products with the same/similar resolutions spanning a range of performance.

A related issue is regarding the benchmark data. It is quite obvious the in-situ data is unevenly distributed in space and also between the regions discussed in the paper. How will the unevenly (and sparsely) distributed snow course data impact the reference data set. And to which extent will that influence the evaluation scores? Further considerations about that would strengthen the paper.

Issues related to sampling bias, data distribution, and SWE measurement type (gamma attenuation measurements or snow courses) are addressed more fully in a companion

paper by Mortimer et al. We originally intended to better coordinate these submissions so that reviewers could be aware of their complementarity in focus, so our apologies for that omission. We have also revised the text to more clearly articulate how our aggregation of the reference data helps to more fairly sample the available distribution of reference data. Finally we note that the scale to which the reference data is aggregated and the regridding of the products does not substantially alter the *relative* assessment of the products. While these decisions do alter the specific values calculated for uRMSE, bias, correlation somewhat, products are still ranked similarly and end up with the same assessed performance overall. This is a reason our overall evaluation strategy focuses on product rankings rather than exact performance measures. In fact our choice to aggregate data within a search radius of 100km tends to improve both the assessed bias and pattern skill for most products. This is demonstrated in the figure below (for mountain regions where this makes a larger difference)[LM1].

[Figure]

*Figure:* Dependence of 2-component skill statistics on aggregation radius (including point-wise comparison=no aggregation). Movement of statistics up and to the left indicates improved performance in both statistics.

Further I miss a reflection on the spatial scale of snow cover in mountainous regions. Performing a comparison on 0.5°x0.5° grids will smooth out the natural variability in complex terrain. I think that should be more thoroughly analysed and discussed.
-As mentioned now both the time varying and climatological tests over NAm use the native product resolutions. While we still aggregate the reference data over 200km search windows (100km aggregation radius) in mountain regions, we argue that this is helping to decrease the influence of sampling bias from the insitu data. From the above figure it is clear that while our choices aggregation scale may affect the absolute statistics to a degree, we are still obtaining well-sorted product rankings which are the key output metric of our analysis.

In the beginning of Chapter 2.3 the paper would benefit from a brief introduction in order to prepare and give the readers an idea of the information presented in the next sections.
-We have revised the text in this section in line with comments from reviewer 1 as well.

Line 49: Mortimer et al. 2024 is missing in the reference list (see also comment to line 290)
Line 290: Mortimer et. al, 2023 is missing in the reference list. (see also comment to line 49. If this refers to the paper referred to as submitted in the references I would recommend to be consistent with the references...)
-This is the companion paper that is also under review. We didn't have a link/DOI to cite originally but have now added a temporary citation to the draft manuscript.

Line 57: The term "authoritative" is pretty ambiguous. I would recommend using a more moderate term ;-)
-This word has been changed to comprehensive.

Table 1: Add a column with original resolutions.
We propose to add the requested column if it can be fit by the journal production while maintaining the same portrait orientation for the table (it not immediately apparent to us if there is sufficient width). If there is insufficient width we would prefer to omit the product resolution since it is a poor explanatory variable. We believe the other information provided is more important and the table is more easily read in its current orientation.

Line 109: Explain IMS.
-Added. Thank you.

Line 177.  The expression  "....method tends to sample" is vague. Please be more specific.
-We have reworded the sentence.

Line 244 (and further lines 487, 489). For consistency, please upcase CCI ( to SnowCCI)
-Changed throughout the paper.

Figure 3 contains a lot of information. For easier interpretation I would recommend to add axis titles in all panels.
-Added.

Figure 6: Please explain the term FM.
-We have now written these out as months.

Line 428 (Chapter 3.3). Please be more consistent with the use of "mountainous" and/or "alpine". Maybe stick to one of them?
-The goal was to stick to mountainous, but we were unsuccessful. Now changed.

Line 438 - 440: Why is that causing these anomalous trends? Please add an explanation.

-It's not fully apparent. Our analysis suggests possible explanations are fluctuations in the availability in situ snow depth data or seasonal/regional variability in the detection of snow presence via passive microwave brightness temperature since both of these are assimilated within the JRA-55 analysis. We have added an additional reference to the JRA-55 paper describing how this information is incorporated in the reanalysis and moderated our claimed attribution to this process slightly.

Line 483-485: Is that really the case in all regions? Justification in a graph similar to fig 1a separated into domains would be a nice supplement.
In this content, the in situ data referred to was not the combined snow course/gamma reference data but rather the weather station snow depths assimilated as part of the GS/CCI retrieval algorithms. We have reworded the text to make this clear.

Line 532- . Here I feel the authors are speculating instead of pointing at real properties of the input data. Lines 532-538 need a second look, and maybe rephrasing in order to be more concrete.
-We have reworked this text to make it more explicit and less speculative and brought in references to relevant prior work.

Line 546: I like it!
-no change needed!

---

## Author Comment (AC3)

Response to Reviewer 1
Responses in blue.

Mudryk et al. aim to evaluate 23 different SWE products based on how well they represent SWE climatology, variability, and trends across mountainous and non-mountainous regions in North America and Eurasia. Using existing and newly created reference datasets, the gridded products are scored using skill target diagrams, resulting in a series of Taylor and target plots, eventually leading to an average ranking of the SWE products. The methodology and technical approach to this evaluation is clear. However, the presentation of the datasets, methodology, and results is notably convoluted or disorganized at times, dampening the impact that this thorough analysis could have. In particular, a clear workflow figure could aid in the introduction of the overall evaluation strategy, where related text often references several other sections, causing for much back-and-forth within the manuscript. While repetition in stating the methodology is appreciated, sometimes the methods, results and associated discussion appear in a single section, making the information challenging to process. Thus, I include no major analysis comments and suggest the authors primarily focus on restructuring the manuscript for a clearer portrayal of meaningful results. My more pointed comments below should help with these review items.

Thank you for the time and effort to review our study and the generally supportive conclusions. To respond to the above comments we have rewritten large parts of Section 2 and 3.3 in order to incorporate your specific comments below (and to try and reduce the amount of back and forth that was noted).

We also note the following 3 analysis changes that affect some of the product-specific statistics, but don't alter generalized rankings or conclusions about product performance:

1. There was an error in the skill score components calculated for Figure 3 that has been fixed (the bias and amplitude values used in the figure were incorrectly normalized, but the relative rankings were still correct).
2. We have altered the way we calculated skill scores for climatological SWE in the NAm region to be more similar to the time-varying statistics and to ensure the statistics reflect the native resolution of the individual products. This is related to a comment from reviewer 2.
3. We added an overall scale factor to both S_bias and S_pattern that allows readers to assess how product performance varies across differing regions/terrain throughout the paper (essentially skill scores for a regional/terrain-specific test are scaled by the uRMSE of the worst performing product across all tests). Since this doesn't alter relative rankings on a particular test it only affects the perceived performance on the given test relative to the other tests. Text in the revised manuscript describes this.

Detailed/line-by-line comments:
Suggest writing out each abbreviation first (e.g., ILAMB, AMBER, IMS)
-Will do - thanks for catching that.

Table 1: Suggest placing a reference column and stating, prior to the table, that each "family" of SWE product will be discussed in more detail following the table and/or placing the text prior to the table.

-We have expanded upon the sentence at line 81 to state this more clearly.

Section 2.2
This section is generally challenging to follow, yet it is intended to set up the pertinent evaluation strategy. There is reference to various sections ahead of the current (e.g., full details referenced in section 2.4, reference to 7 products chosen for an ensemble). Perhaps a schematic of the workflow/evaluation scheme would be helpful. As such, it is also unclear when a point system was introduced (line 153-154).

-We have rewritten this section discussing the point system up front and more explicitly and tried to remove some of the back and forth you mention.

Line 156: Unclear what exactly expert judgement is considered in this case

-We have reworded this section and this phrase no longer appears.

Table 2: Suggest including some justification as to how regions were selected. Unclear at what spatial scale these variables are evaluated.

-The rewritten intro to Section 2.2 now states up front that most regions were selected based on the characteristics of the reference data. We have also added more detailed rationales when discussing the reference data in Section 2.3.

Line 172: It would benefit the readership if the text stated the spatial extent in addition to the figure.

-The available coverage over both continents is explicitly stated in the revised text.

Figure 2 c-d: It is unclear what the scale bar is referencing, as it pertains to climatology. If this is "SWE climatology," or peak SWE magnitude, labeling it and stating so in the figure caption, as opposed to "climatology," would be helpful.

-only the nonmountainous climatology is shown now and it has been labelled as "Nonmountainous SWE Climatology (Bias-corrected GlobSnow v3)".

Line 219: It is unclear what "broad a range of meteorological analysis fields as possible" looks like for the 7 selected products, which are spelled out later in this paragraph. Could an example be provided compared to a product that was not selected?
Line 221: Consider removing the following sentence, as it again jumps ahead to several sections from the current and causes confusion: "It will be demonstrated in Sect. 4 that the reanalysis-type products which employ assimilation of surface snow information all have seasonal incongruities with one another." Suggest revising line 124 for similar reasons.

-The rewritten section 2.3 addresses both of these comments (L219 and 221).

Section 2.4: Please number and format equations similarly, as there are many and some build off of one another. Reference to a figure here or in one of the citations may bring additional intuition to this methodology here, prior to seeing the results.

-Equations have been numbered, and we have moved some of the subsequent description on how target diagrams display information to the start of the paragraph.

Line 249: Suggest a ½ to full sentence on why this approach was taken to rank similarity across products (were there other approaches in consideration?).

-The advantages of the two-component skill scores we use compared to uRMSE (which is what is used in Taylor diagrams --- another typical approach) are explained in the following sentences of the paragraph. They are also contrasted in the results presented in Figure 3.

Line 353: Additional annotations on this figure would be helpful. For example, placing notation near Crocus-ERA5 and ERA5-Land on the upper right panel would aid in the necessary scanning between text and figure (especially since the numbers/rankings obviously change between panels). This comment extends to Figure 4 and 5. Suggest also reiterating what is represented on each axis in each figure and/or across panels, particularly in the Taylor plots and in reference to pattern statistics.

-In place of additional notation on an already complex figure, when discussing Figure 3 we have specified the rankings of Crocus-ERA5 and ERA5-Land on the taylor plot to make it easier for readers to identify their positions. We also note the statistics of the two products are better separated (thus easier to read) on the figure using the revised method for assessing product SWE climatologies in mountainous regions.

Section 3.3: Many of these beginning sentences/paragraphs, aside from the sentences explicitly
referring to Figure 7, read as though they belong in the methods or discussion sections, which decreases the impact of the following results. There is a lot of information to unpack in Figure 7. The paragraph structure for each row is appreciated. Perhaps the authors would consider beginning each paragraph with the intended take home point, particularly for the bottom row of results.

-As suggested we have removed much of the methods-related preamble and front-loaded a take-home point in the majority of the paragraphs. We also simplified the messaging regarding the EO-trends (bottom row of results).

Section 4: The final result, presented quite clearly, does seem as though it could live in the results section. Are there comparable results specific to the select regions (NA and Eu mountain and non-mountain)? This question also pertains to my next comment.

-We elected to place the final product rankings in the discussion section because it distinguishes it from the more complex and varied individual results detailed in Section 3 (thereby highlighting it we would argue) and because in discussion of the figure and the overall results leads directly into discussion-appropriate commentary.

-Regarding regionally specific results, to an extent the peak season results presented in Figure 5 partially fulfills this function. But since the full suite of tests does not use the same choice of regions for all tests (for the reasons outlined in Section 2) it's not simple to compare the regions you mention in your comment in a meaningful way. For example, because the performance in mountain regions (which are evaluated over NA only) is a key differentiator of performance, NA-only rankings would be similar to the final rankings. EU-only rankings would still be able to distinguish two distinct product groups (apparent from examining Figures 3-nonNH and 5-EU,): the four GLDAS products, JRA-55, JAXA and ERAint-Land are consistently in the bottom half of the distributions distinct from the remaining products. But the EU-only test would be unable to differentiate as much among the top products because their performance is similar to one another as far as we have available reference data over the region to assess.

Line 535: Can the authors expand and be more explicit about "The relative overall rankings shown in Fig. 8 are meant to function as a guideline only"? The following conclusion also states that "user needs and objectives must guide product selection," however a lot of technical and thorough work went into the culminating Figure 8. Are there thus product recommendations and takeaways for users broadly and by region?
-We have expanded upon this statement in the revised manuscript. We stand by our results to the degree that our coverage of in situ data permits us to generalize. But this statement was meant to acknowledge that some products can have idiosyncratic regional performance. For example, the GLDASv21 performance assessed only over the CONUS (Figure S2) performs much better, especially in CONUS mountainous terrain where it's ranked 4[th], compared its overall NAm performance (ranked 14[th]) and its overall ranking (18[th]). This is why we have provided the caveat about rankings functioning as a guideline for hemispheric performance, but that for specific regions there may be differences.
Likewise we realize that the absence of reference data from mountainous regions of Europe and western Asia is a clear gap in our ability to assess any deficiencies over these regions that aren't also reflective of the products' performance over North American mountainous regions.

Discussion: There lacks a discussion on limitations to this assessment and consideration of other gridded SWE products
Limitations of the assessment related to the reference data distribution were stated on L555-558. Limitations related to the use of distributions to assign scores were mentioned in L560-562. Beyond this, we are unsure what your comment on consideration of other gridded SWE products means. While we have not evaluated every gridded SWE product that there is, the results provide a general procedure to which additional products could be incorporated. We do state this explicitly now in the new text.

---

## Author Response (AR2)

I made the requested change at line 182: 'evaluated' to 'evaluate'

Please note I found one additional error in supplementary figure 2. This figure was originally used to provide evidence for two statements in the main text regarding the performance of GLDAS v2.1. There is still analogous evidence for the original statement in a figure from the main document (copied below). I've therefore altered two sentences that originally referred to the supplementary figure to refer to the main text figure instead.

Original statement 1 (with context):
*"All four GLDAS products perform very poorly when evaluated hemispherically (Fig. 5) due in part to large biases (Figs. 2 and 4). However, GLDAS v2.1, which uses different precipitation forcing than the other three versions, performs better when evaluated regionally over the continental United States (Fig. S2), especially in mountainous terrain. Thus, while it is tempting to extrapolate regional performance, this product provides a good counter-example where doing is particularly detrimental."*

New text:
"In general, the GLDAS products perform poorly when evaluated hemispherically (Fig. 5) due in part to large biases (Figs. 2 and 4). However, GLDAS v2.1, which uses different precipitation forcing than the other three versions, performs better when evaluated over nonmountainous North America (Fig. 5). Thus, while it is tempting to extrapolate regional performance, this product provides a good counter-example where doing is particularly detrimental. "

Original statement 2:
"GLDAS v2.1 provides a specific example where its performance over the continental US does not reflect its much poorer performance outside that region."

New text:
"GLDAS v2.1 provides a specific example where its performance over nonmountainous North America does not reflect its much poorer performance outside that region."

[Figure]

Figure 5 (as before, now used to justify statements above)

[Figure]

Revised figure S2.